# Healthcare Providers’ Knowledge of Value-Based Care in Germany: An Adapted, Mixed-Methods Approach

**DOI:** 10.3390/ijerph19148466

**Published:** 2022-07-11

**Authors:** Axel Kuck, Kristian Kinscher, Leonard Fehring, Helmut Hildebrandt, Johannes Doerner, Jonas Lange, Hubert Truebel, Philip Boehme, Celina Bade, Thomas Mondritzki

**Affiliations:** 1Faculty of Health, School of Medicine, Witten/Herdecke University, 58455 Witten, Germany; axel.kuck@uni-wh.de (A.K.); kristian.kinscher@gmail.com (K.K.); leonard.fehring@uni-wh.de (L.F.); johannes.doerner@t-online.de (J.D.); htruebel@gmx.de (H.T.); philip.boehme@uni-wh.de (P.B.); celina.bade@uni-wh.de (C.B.); 2Optimedis AG, 20095 Hamburg, Germany; h.hildebrandt@optimedis.de; 3Department of Surgery II, Helios University Hospital Wuppertal, 42283 Wuppertal, Germany; 4Cologne-Merheim Medical Center, Department of Abdominal, Vascular and Transplant Surgery, Witten/Herdecke University, 58455 Witten, Germany; langej@kliniken-koeln.de; 5Bayer AG, Cardiovascular Precision Medicine, 42096 Wuppertal, Germany

**Keywords:** value-based care, cycle of care, value-based reimbursement, health reform

## Abstract

Background: Value-Based Care (VBC) is being discussed to provide better outcomes to patients, with an aim to reimburse healthcare providers (HCPs) based on the quality of care they deliver. Little is known about German HCPs’ knowledge of VBC. This study aims to investigate the knowledge of HCPs of VBC and to identify potential needs for further education toward implementation of VBC in Germany. Methods: For evidence generation, we performed a literature search and conducted an online survey among HCPs at 89 hospitals across Germany. The questionnaire was based on published evidence and co-developed with an expert panel using a mixed methods approach. Results: We found HCPs to believe that VBC is more applicable in surgery than internal medicine and that well-defined cycles of care are essential for its application. HCPs believe that VBC can reduce health care costs significantly. However, they also assume that implementing VBC will be challenging. Conclusions: The concept in general is well perceived, however, HCPs do not want to participate in any financial risk sharing. Installing an authority/independent agency that measures achieved value, digital transformation, and that improves the transition between the inpatient and the outpatient sectors are top interests of HCPs.

## 1. Introduction

Health systems worldwide struggle with rising costs [1]. Currently, most health systems in Europe use a fee-for-service (FFS) reimbursement system [2]. In such a system, every single healthcare procedure or service is reimbursed as an isolated event according to fee schedules. In recent years, this system has been increasingly criticized for being agnostic to the quality of the outcomes of medical services and associated with “unnecessary care” and redundancies in the provided services [2,3]. Michael E. Porter outlined the concept of value-based care (VBC) in a seminal article in 2010 [4]. A focus of Porter’s research was in particular the coming of age of VBC in Germany [5]. The cornerstone of the concept is holistic, value-driven competition described with the formula Value = Health Outcomes/Costs. Thus, value can be generated by improving health outcomes and/or reducing costs. VBC aims to incentivize all stakeholders (e.g., HCPs such as physicians) in the health system to work toward value and to penalize non-conformity (e.g., performing unnecessary procedures). Thus, VBC also incentivizes providers to maximize the difference between patient health benefits and the total cost incurred for a service [6]. Due to the necessity to measure outcomes and costs in VBC, the concept is closely linked to evidence-based medicine, which enables the translation of medical evidence to value measures, such as improvement of quality of life, through an integration of research evidence, clinical expertise, and patient values [7]. To assess the value, the consideration of cycles of care can be helpful. They describe the entire course of treatment of a patient from the first contact with the health system (both the in- and outpatient sectors) to a pre-specified end point (e.g., “cure”). In Germany, steps have recently been taken toward the implementation of VBC [8,9,10]. An example is the German Pharmaceutical Market Restructuring Act (Arzneimittelmarktneuordnungsgesetz, or AMNOG) that was introduced in 2011. It was broadly regarded as successful, and it ties the price of a prescription drug to a holistic value-based benefit assessment in the first year of its market availability [11]. Another recent initiative is the Digital Care Act that commits insurance payers to refund patient-centered digital healthcare applications (Digitale Gesundheitsanwendungen, DiGA) based on provided value [12]. Although HCPs are critical for the implementation of VBC [5], little is known about the knowledge and the perception of German HCPs on VBC. Therefore, we sought to investigate causes, hurdles, and chances for the further implementation of VBC via HCPs as a key interest group. Hence, relevant literature was identified, and an online survey was performed as part of a mixed-methods approach. This study thus aims to investigate the knowledge of HCPs on the dimension of VBC and to identify potential needs for further education toward implementation of VBC in Germany. By doing so, the study delivers an important contribution to the academic literature as it closes the gap in understanding the perspectives of German HCPs on VBC and creates transparency on pain points that need to be considered in future VBC related policies. In line with the methodology and its limitations, we put our findings in perspective and provide suggestions for further reading.

## 2. Materials and Methods

A comprehensive literature search was performed for articles published in print or online up until 27 October 2019 from the search engine PubMed^®^ of the United States National Library of Medicine using the search strategy “(value-based care [Title/Abstract]) OR value-based health care [Title/Abstract]) OR vbhc [Title/Abstract])”. All articles in English or in German with a study aim clearly related to VBC were included in the analysis. The search engine PubMed^®^ was selected as there is no similar platform specifically for the German context. Generally, German publications in the field are also represented within this search engine. Results were analyzed to cluster fields of research, country of origin, medical specialty, and year of publication.

The literature review results were synthesized to build a first view on focus themes. The derived focus themes were the payment model (e.g., fee-for-service), differences in the applicability of VBC depending on the concerned medical field, cycle of care, costs and resources, risk participation of HCPs, and digital transformation, as well as the relevance of VBC from the perspective of HCPs. An expert panel consisting of 10 members with multi-professional backgrounds in medicine, pharmaceutical research, psychology, and economics generated survey questions based on this synthesis. The majority of the initially 80 generated questions resulted from the focus themes found in the literature review. Additionally, the expert panel tried to include a meta-perspective on VBC based on the literature review findings. In an iterative discussion process, 62 questions were deprioritized, and 18 were included in the final survey. The questions address Porter’s formula of VBC [4], encompassing the areas of value, result, and cost. In addition, aspects of implementation that are particularly relevant to the German health system (e.g., transition between the inpatient and the outpatient sectors) are covered. The overall study design is outlined in Figure 1.

Finally, questions were rephrased as Likert items with an answer range from 1 (=strongly disagree) to 5 (=strongly agree); all 18 items can be seen in Figures 4–7. In further analysis, we regarded ratings > 3 as an agreement.

As recommended by PsyWeb criteria [13], the survey was pre-tested prior to launch. The online survey was conducted with LimeSurvey© Version 3.23.7 + 201006 (LimeSurvey GmbH, Hamburg, Germany). HCPs were recruited by sending the survey link to the HCP mailing list of a large, German private hospital chain, which covers all clinic sizes and thus the entire spectrum of inpatient care in Germany (89 German university and non-university hospitals were included). Final year medical students were contacted via student mailing lists. At the beginning of the online survey, information about the general concept of VBC was displayed to participants including external references to ensure an equal understanding of VBC. Subsequently, participants were asked to rate the 18 Likert items.

Descriptive statistical analysis was performed using Microsoft Excel© (Microsoft Corporation, Redmond, WA, USA) and IBM SPSS Statistics 27© (IBM, Armonk, NY, USA).

## 3. Results

### 3.1. Identified Published Studies for Evidence and Hypothesis Generation

Search results yielded 897 references. Because they did not meet the inclusion criteria, 341 publications were excluded, leaving 556 for analysis. We found that 82.7% of the included publications originate from the United States, with 6.1% from Europe. The majority refer to surgery (35.1%), followed by internal medicine (19.8%). Within surgery, trauma and orthopedic surgery stand out as the leading specialties (64.1%), followed by neurosurgery (9.7%). Within internal medicine, hematology and oncology are the most prominent (27.3%), followed by cardiology (22.7%) and gastroenterology (20.0%). Especially in Europe, the amount of research was still very limited in the field at the time when the literature review was conducted. However, numbers of publications in general per year increased significantly within the last 10 years, as shown in Figure 2. To the authors’ best knowledge, no survey similar to that presented here was conducted in the German context in any other study found in the literature review.

### 3.2. Online Survey

Overall, 1004 HCPs participated in the survey. In total, 317 participants were excluded due to reasons displayed in Figure 3.

The 687 participants reported being: physicians (438), nurses (65), other profession (58), and final year medical students (126). A total of 51.4% of participants reported being male, and 46.4% reported being female. Excluding medical students, 36.7% of participants reported having a leading position within their institution. Further demographic information is depicted in Table 1.

### 3.3. Although Regarded as Desirable, VBC Is Not Yet Broadly Implemented in Clinical Practice

According to our survey, VBC is considered a relevant concept for a health system of the future in the eyes of the participants (M = 3.89, SD = 0.98; 70.9%; MED = 4), with a level of agreement of 70.9% (Figure 4A). In contrast, participants broadly disagreed that VBC is a relevant concept in their institution (M = 2.60, SD = 1.24; 25.7%; MED = 3) (Figure 4B). There is a broad consensus that the current fee-for-service model incentivizes physicians to offer procedures that are not necessary (M = 4.00, SD = 1.03; 72.5%; MED = 4) (Figure 4C).

### 3.4. VBC Can Be Implemented across Medical Fields, but the Cycle of Care Needs to Be Well Defined

Generally, VBC is regarded as applicable in internal medicine (M = 3.35, SD = 1.04; MED = 3) as well as in surgery (M = 3.80, SD = 0.98; MED = 4) (Figure 5A,B). However, the level of agreement was higher for surgery than internal medicine (71.1% vs. 47.0%). In a well-defined cycle of care, application of VBC was rated as superior to the not well-defined (M = 3.74, SD = 0.90; 65.2%; MED = 4 vs. M = 2.38, SD = 1.02; 15.0%; MED = 2) (Figure 5C,D).

### 3.5. VBC Is Regarded as Carrying Significant Potential to Reduce Costs and to Increase Efficiency

Participants agree that VBC will contribute to a more efficient utilization of resources within the health system (M = 3.81, SD = 1.01; 68.2%; MED = 4) (Figure 6A). Participants also believe that VBC has the potential to reduce procedures that come with a low success rate (M = 3.71, SD = 1.04; 64.3%; MED = 4) (Figure 6B). Additionally, participants believe that VBC has the potential to save costs in general (M = 3.71, SD = 0.99; 64.8%; MED = 4), albeit less in administration (M = 3.07, SD = 1.24; 39.9%; MED = 3) (Figure 6C,D).

### 3.6. Although the Implementation of VBC Is Seen as Difficult, Clear Expectations from HCPs Exist

In general, participants do not expect the implementation of VBC to be easy in the current health system (M = 2.70, SD = 1.02; 21.4%; MED = 3) (Figure 7A). Participants did not agree that physicians should share the financial risk if their procedures are unsuccessful (M = 2.45, SD = 1.22; 21.1%; MED = 2) (Figure 7B). In this regard, final year medical students responded to be more likely to share financial risk for unsuccessful procedures than all other professions (Mann-Whitney U = 29,328.00, Z = −1.993, *p* = 0.046 two-tailed). Participants agreed that “pharmaceutical companies should share the financial risk if a drug does not work.” (M = 3.68, SD = 1.24; 63.1%; MED = 4) (Figure 7C). However, concern remains that the implementation of VBC will hinder scientific progress (M = 3.52, SD = 1.11; 55.2%; MED = 4) (Figure 7D). Participants also did not share the opinion that “an authority to rate the value of procedures/curative treatments can easily be installed.” (M = 2.95, SD = 1.06; 32.2%; MED = 3) (Figure 7E). Concerning the ecosystem of VBC, our results suggest disagreement on that the interface between the inpatient and the outpatient sectors is no hurdle for the successful implementation of VBC (M = 2.68, SD = 1.19; 27.3%; MED = 2) (Figure 7F). Furthermore, a digital transformation in the health system was considered to be a requirement for the success of VBC (M = 3.63, SD = 1.10; 58.1%; MED = 4) (Figure 7G). Interestingly, results of Spearman correlation indicated that there was a positive association between the age of the participants and agreement to necessity of digital transformation, (rs (659) = 0.09, *p* = 0.0021), meaning that the higher the age, the higher the agreement. No other significant relationships of gender or age on participant responses was found.

## 4. Discussion

We found that research on VBC has increased significantly over the last decade. A driving factor could be the increasing pressure on health systems with rising costs, e.g., higher procedure volumes in surgery that are not necessarily in combination with better outcomes [14]. Furthermore, digitalization in general as well as improved accessibility and availability of digital outcome measurement devices and thus more precise outcome measures (e.g., sensors measuring real world walking speed) could have led to an increase in VBC research, as these developments facilitate comparing costs to outcomes [15,16]. The German government has recently taken several measures to simplify the use of digital tools with the Digital Healthcare Act (Digitale-Versorgung-Gesetz, DVG), which might nurture the implementation of VBC in Germany [17]. However, most of the research is still performed in the United States, which could be explained by the exceptionally high costs in the US health system, as shown by Papanicolas and colleagues [18], and the less strict restrictions regarding health data protection as a prerequisite for the measurement of outcomes [19,20].

We performed a survey on VBC on the largest German HCP cohort so far. Overall, our findings show that VBC is seen as relevant for a future health system in Germany, which is in line with research by Porter and Guth [5]. However, HCPs state that VBC is not a relevant concept within their institution. This was also shown by previous research, which hypothesizes that this is due to the fragmented system and potentially the lack of continuity of care between the outpatient and the inpatient sectors, which is particularly pronounced in Germany [21,22]. An alarming finding is that 72.5% of HCPs currently see incentives that encourage them to offer unnecessary procedures. This clearly questions the current fee-for-service model and the lacking continuum of care.

We found that HCPs consider VBC to be slightly more applicable in surgery than internal medicine. This is in line with the literature identified, and it is potentially driven by the broader availability of cycles of care and the accessibility of outcome parameters. Additionally, there are numerous, currently ongoing efforts in the development of digital biomarkers to measure outcomes, e.g., mobilized-d [23,24]. Furthermore, we found that the cycle of care is regarded to play a crucial role to achieve a transition to VBC. They need to be mapped and transparent to ensure that outcomes are measured in an objective and a comparable manner [25,26].

A prominent finding in our study is that VBC is thought to improve efficiency and drive down costs. Furthermore, the perception of HCPs is that the number of procedures with low success rates will be lowered by VBC, which potentially reduces overall costs. While this is economically desirable, from the perspective of an individual patient this may lead to limitation of treatment options, as seen for example in countries such as the UK [27]. However, there is no strong believe that VBC will lower administrative healthcare costs, which are particularly high in Germany, e.g., with still more than a hundred salutary health insurance plans and the fragmentation of the system in general [28,29]. Overall, HCPs believe in cost reduction through VBC.

HCPs agree that the transition to VBC is challenging. This finding is in line with former research that found that organizational change is especially difficult in the health sector [30,31]. Despite the overall positive perception of VBC, HCPs think that financial risks due to outcome-based reimbursement should not be carried out on the shoulders of physicians. A hypothesis could be the fear of loss of income due to novel VBC-inspired payment models (e.g., potential penalties). In contrast, HCPs believe that pharmaceutical companies should be held responsible if their medication does not deliver promised effects. HCPs believe that if innovations with only small additional benefits are no longer reimbursed, scientific progress will suffer. This finding is interesting since many novel medications result in only modest clinical improvement [32].

Furthermore, mild disagreement resulted regarding the establishment of an authority/an institution that evaluates the value of healthcare interventions or treatments (Figure 7E). In Germany, the Institut für Qualität und Wirtschaftlichkeit im Gesundheitswesen (IQWiG, Institute for Quality and Efficiency in Health Care) was founded in 2004; although one of its tasks is to generally evaluate medical procedures, it lacks the authority to determine whether a concrete carried out procedure will be reimbursed, which might explain the response of the HCPs [33]. In Germany, HCPs see the fragmentation in the inpatient and the outpatient sectors as a hurdle for the implementation of VBC, while a smooth transition is critical to establish an end-to-end cycle of care and outcome measurement [5]. Interestingly, HCPs believe that a digital transformation in the health system is a necessary requirement for the success of VBC. This might be related to improved measurement of outcome parameters and accessibility of health care data, as discussed before [34]. Taken together, HCPs seek: an authority that evaluates achieved value; digital transformation; and an improved transition between the inpatient and the outpatient sectors to advance VBC.

We are aware that the study at hand has certain limitations. Most importantly, not all potentially interested HCPs had access to the survey due to the send-out of the online survey to a pre-selected group. Most of the survey participants were medical doctors with an interest in VBC, which limits the generalizability of our findings to all HCPs. Generalizability is further limited as no inference analysis has been conducted. Furthermore, the sample of participants displays an underrepresentation of older age groups, which may be due to the lower affinity to online surveys of certain age groups or due to organizational and hierarchical structures within hospitals. This may lead to skewed results. Furthermore, the choice of a primarily quantitative online survey based on closed questions cannot fully gather and appreciate potential qualitative aspects of the topic. Another limitation of the study at hand is that the validity or reliability of the questionnaire were not tested.

## 5. Conclusions

In our literature search, we found that (a) most publications originate from the US, (b) most publications are linked to surgery, and (c) a strong increase in research activity took place within the recent decade. Findings from our one-time online survey suggest that HCPs deem VBC to be of relevance regarding it’s potential to save costs and its potential to improve quality and productivity in the healthcare ecosystem. German HCPs judge VBC to be beneficial to the health system in general, but they also recognize hurdles regarding the payment model in their ecosystem. Several topics are on HCPs minds in this context. They especially think about the certainty that costs will be covered by statuary health insurance, clear and well-defined standard processes and operating procedures for diseases, regulatory oversight (e.g., a competent VBC authority), and digital technology (e.g., mobile apps or IT software) to support the transformation of the healthcare market and to counter its fragmentation (e.g., between the inpatient and the outpatient sectors). The present study thus contributed to the existing academic literature by developing an understanding of the perspectives of German HCPs on VBC and creating transparency on pain points that need to be considered in future VBC related policies.

## Figures and Tables

**Figure 1 ijerph-19-08466-f001:**
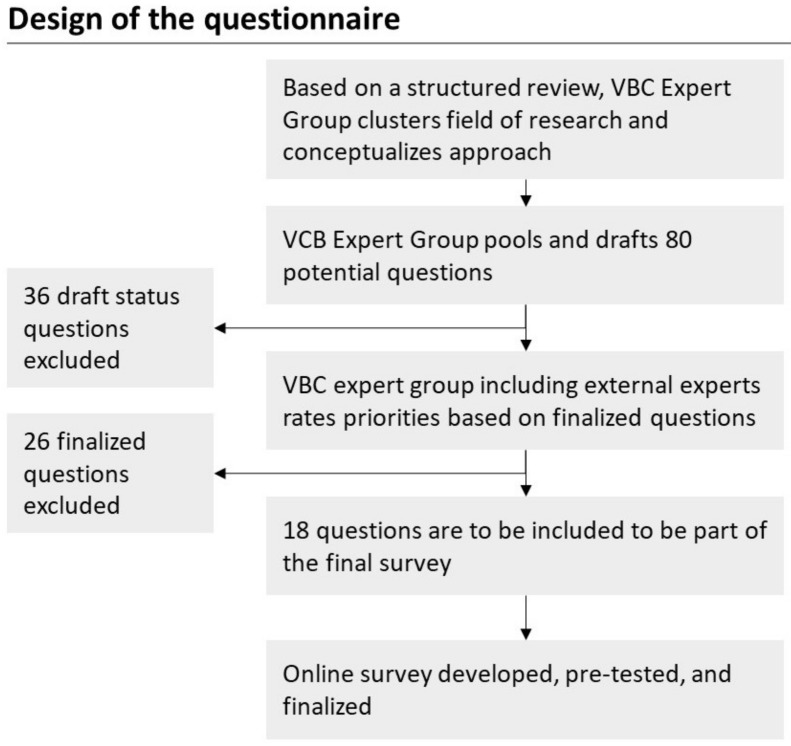
Overall study design.

**Figure 2 ijerph-19-08466-f002:**
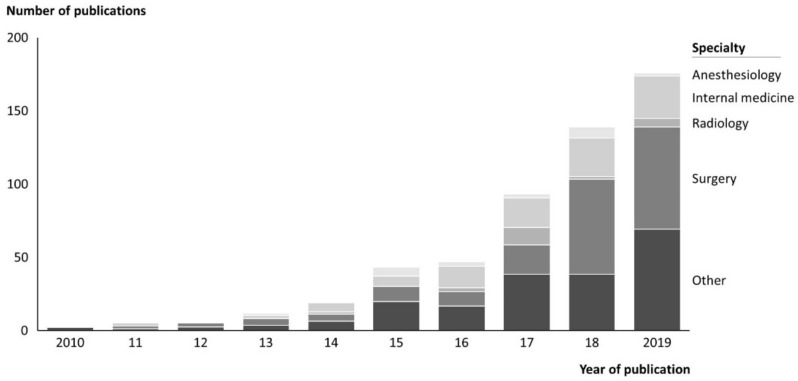
Publications on VBC. The increasing number of VBC related publications are shown over the past 10 years, split by subject.

**Figure 3 ijerph-19-08466-f003:**
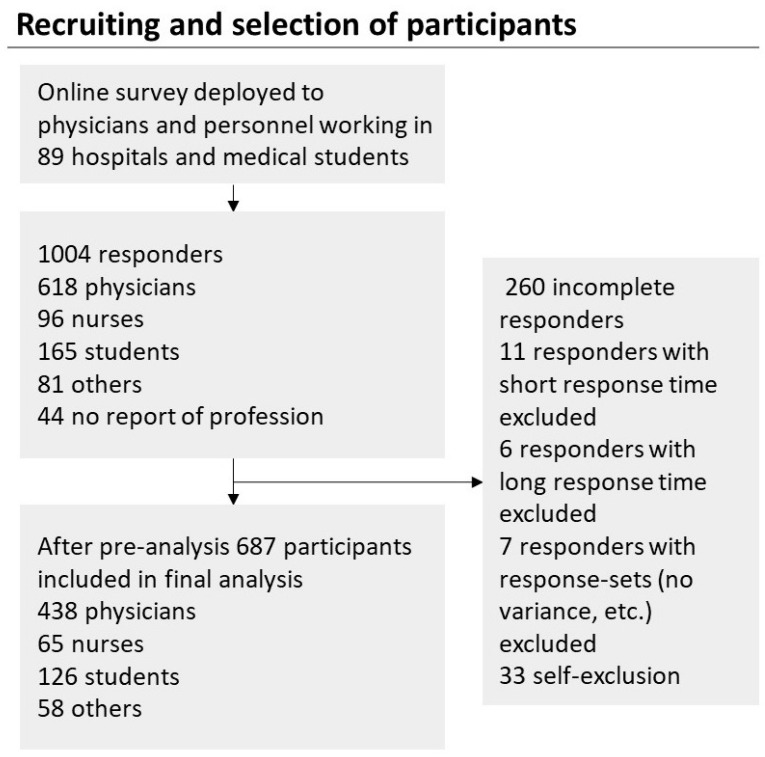
Flow of survey participants, including participants that needed to be excluded due to limited data quality.

**Figure 4 ijerph-19-08466-f004:**
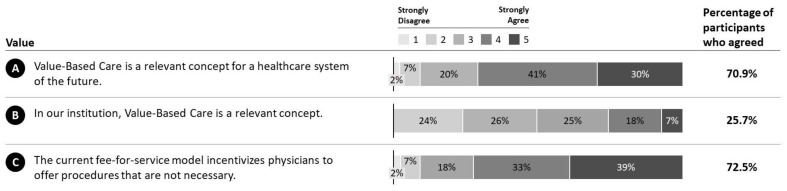
Category value. Responses of participants along for the VBC category value. **A**: Relevance of VBC for the healthcare system. **B**: Relevance if VBC for own institution. **C**: Incentive for unnecessary procedures through fee-for-service model.

**Figure 5 ijerph-19-08466-f005:**
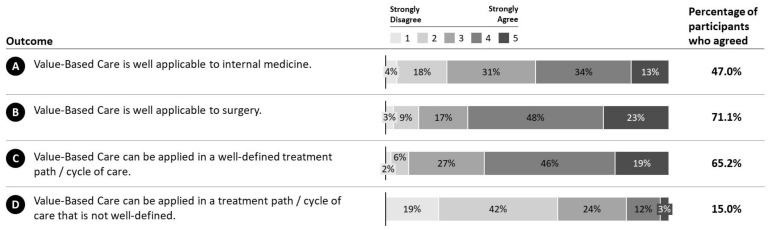
Category outcome. Responses of participants along the VBC category outcome. **A**: VBC applicability to internal medicine. **B**: VBC applicability to Surgery. **C**: VBC applicability to well-defined cycle of care. **D**: VBC applicability to not well-defined cycle of care.

**Figure 6 ijerph-19-08466-f006:**
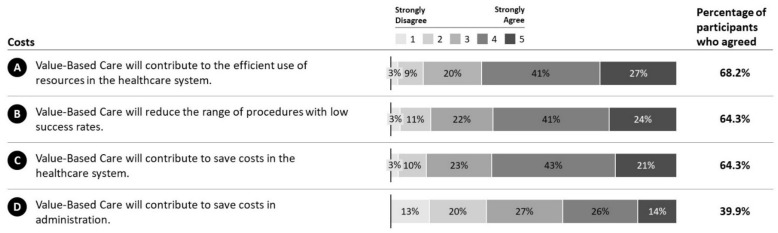
Category costs. Responses of participants along the VBC category costs. **A**: VBC contribution to resource efficiency. **B**: VBC impact on procedure utilization. **C**: VBC contribution to healthcare cost savings. **D**: VBC contribution to administrative cost savings.

**Figure 7 ijerph-19-08466-f007:**
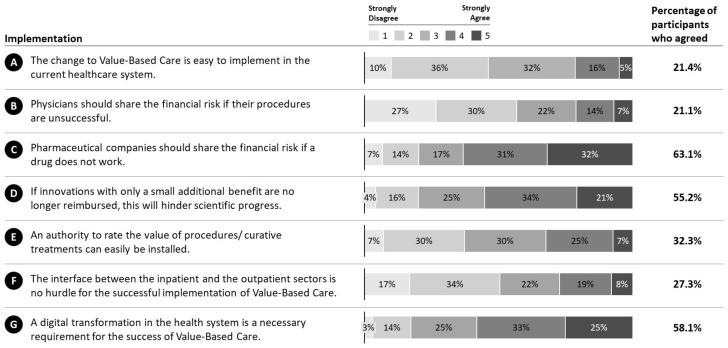
Implementation of VBC. Responses of participants regarding the further implementation of VBC in Germany. **A**: Ease of VBC implementation. **B**: Financial risk sharing of physicians. **C**: Financial risk sharing of pharmaceutical companies. **D**: VBC impact on scientific progress. **E**: Installation of an authority to rate value. **F**: Impact of interface of in- and outpatient sectors on VBC implementation. **G**: Necessity of digital transformation for VBC success.

**Table 1 ijerph-19-08466-t001:** Demographic information of HCPs who participated in the survey.

Demographics	Options	N	Percentage
Gender	Male	353	51.4
	Female	319	46.4
	Divers	1	0.1
	N/A	14	2.1
Age	16–20	12	1.7
	21–30	211	30.7
	31–40	215	31.3
	41–50	110	16
	51–60	110	16
	61–70	25	3.6
	N/A	4	0.6
Profession	Physician	438	63.8
	Nurse	65	9.5
	Medical student	126	18.3
	Miscellaneous	58	8.4
Professional experience (years)	N/A & Med-students	134	19.5
	1–5	156	22.7
	6–10	120	17.5
	11–20	124	18.1
	21–30	106	15.4
	>30	47	6.8
Leading position	Yes	252	36.7
	No	333	48.5
	N/MA	93	13.5
	N/A	9	1.3

## Data Availability

Not applicable.

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
