# Peer review of "Healthcare Providers’ Knowledge of Value-Based Care in Germany: An Adapted, Mixed-Methods Approach"

_ijerph, 2022, doi:10.3390/ijerph19148466_

Round 1
Reviewer 1 Report
The work has a promisable capacity in respect of health care performance, effectiveness and efficiency. There is a lack of information about relationships between participant demographics and responses to questions (e.g. gender). Probably, there was no statistically significant relationships but it would be informative to indicate it.
Author Response
Comment | Implementated Ammendments |
There is a lack of information about relationships between participant demographics and responses to questions (e.g. gender). Probably, there was no statistically significant relationships but it would be informative to indicate it. | We added the following in line 198 -199: No other significant relationships of gender or age on participant responses was found. |
English language and style are fine/minor spell check required | A native speaker went through the text once more and made adoptions where appropriate |

Reviewer 2 Report
The concept of valued based care needs additional clarification based on the literature. It is also important to explain its connection to the concept of evidence-based medicine (EBM). I suggest the reading the following seminal papers:
Conrad, D. A. (2015). The theory of value‐based payment incentives and their application to health care. Health Services Research, 50, 2057-2089.
Brown, M. M., Brown, G. C., Sharma, S., & Landy, J. (2003). Health care economic analyses and value-based medicine. Survey of ophthalmology, 48(2), 204-223.
The authors must discuss the limitations of the study, namely those limitations regarding the online survey. Moreover, it is quite acceptable that respondents’ opinion depends on their expertise in medicine.
Some text needs to be rephrased. For example: “Reimbursement-security, well-defined cycles of care, and regulatory oversight (e.g., a competent VBC authority), as well as digital means to support the transformation of the healthcare market as well as its improvement regarding its fragmentation (e.g., between inpatient and outpatient sectors), are on HCPs minds in this context.” (line 243-246)
Author Response
Comment | Implementated Ammendments |
The concept of valued based care needs additional clarification based on the literature. It is also important to explain its connection to the concept of evidence-based medicine (EBM). I suggest the reading the following seminal papers: Conrad, D. A. (2015). The theory of value‐based payment incentives and their application to health care. Health Services Research, 50, 2057-2089. Brown, M. M., Brown, G. C., Sharma, S., & Landy, J. (2003). Health care economic analyses and value-based medicine. Survey of ophthalmology, 48(2), 204-223. |
We added the following paragraph to the introduction based on the recommended papers and also cited these accordingly (lines 50 to 55, citations now as 6 and 7): "VBC thus also incentivizes providers to maximize the difference between patient health benefits and the total cost incurred for a service [6]. Due to the necessity to measure outcomes and costs in VBC, the concept is closely linked to evidence-based medicine. Which enables the translation of medical evidence to value measures such as improvement of quality of life through an integration of research evidence, clinical expertise and patient values and [7]." |
The authors must discuss the limitations of the study, namely those limitations regarding the online survey. Moreover, it is quite acceptable that respondents’ opinion depends on their expertise in medicine. | We added the following paragraph to the discussion (lines 269 to 279):We are aware that the study at hand has certain limitations. Most importantly due to the send-out of the online survey to a pre-selected group, not all potentially in-terested HCPs had access to the survey. Most of the survey participants were medical doctors with an interest in VBC which limits the generalizability of our findings to all HCPs. Generalizability is further limited as no inference analysis has been conducted. Furthermore, the sample of participants displays an underrepresentation of older age groups, which maybe due to the lower affinity to online surveys of certain age groups or due to organizational and hierarchical structures within hospitals. This may lead to skewed results. The choice of a primarily quantitative online survey based on closed questions can furthermore not fully gather and appreciate potential qualitative as-pects of the topic. Another limitation of the study at hand is, that the validity or relia-bility of the questionnaire were not tested |
Some text needs to be rephrased. For example: “Reimbursement-security, well-defined cycles of care, and regulatory oversight (e.g., a competent VBC authority), as well as digital means to support the transformation of the healthcare market as well as its improvement regarding its fragmentation (e.g., between inpatient and outpatient sectors), are on HCPs minds in this context.” (line 243-246) | We rephrased to "Several topics are on HCPs minds in this context. Especially they think about the cer-tainty that costs will be covered by statuary health insurance, clear and well defined standard processes and operating procedures for diseases, regulatory oversight (e.g., a competent VBC authority), and digital technology (e.g. mobile apps or IT software) to support the transformation of the healthcare market and counter its fragmentation (e.g., between inpatient and outpatient sectors)" (now line 287-292) |

Reviewer 3 Report
Regarding the review of the paper entitled "Healthcare Provider´s Knowledge on Value-Based Care in Germany, an Adapted Mixed Methods Approach", I have the next few suggestions:
1) The content of the paper shows a link with the title.
2) The Abstract provides all the needed information.
3) The aim of the paper should be presented in the Introduction (be consequent, formulate it as it is done in the Abstract).
4) Line 44: Please use firstly the full name (Value Based Care) in the Introduction and put the abbreviation in brackets (VBC), as it is done in line 21 (Abstract); then you can use the abbreviation in the rest of the paper with no need to explain it.
5) The theoretical background of the study is not very strong (presented only in the Introduction), please provide additional theoretical support to this study. It could constitute a separate section Literature review/ Literature background. You should include a literature review section after the Introduction and consider formulating research questions or hypotheses.
6) Research methods – some additional information about the methodological procedures is necessary, specifically about the hospitals chosen by the Author(s). It is written: “HCPs were contacted directly via their email or their respective intranet (89 German University and non-University hospitals were included)”. Just few phrases would be recommended to characteristic these hospitals. What was the criteria of choosing these 89 hospitals?
7) An online survey is appropriately designed and the results are well presented.
8) Maybe because the literature search was performed for articles from the search engine PubMed® of the United States National Library of Medicine, 82.7% of the publications originate from the United States, and only 6.1% from Europe. Perhaps as the paper concerns Germany, it would be reasonable to perform the literature search using another abstract and citation database of peer-reviewed literature. The literature study is quite flat and limited. This section does not bring the reader to a clear state of the art of the literature on this topic. It would be interesting to read about research already conducted in Germany and related to this area. What about other countries? Is it somewhere already done (similar surveys)? Are 556 articles theoretical ones or empirical ones? It would be interesting to know if they present the global perspective or specific solutions for a precise country or even for one HCP (case study), etc. The characteristic of selected articles used in the study is missing.
9) The Authors must justify that they truly fill a gap (theoretical / methodological / empirical) in the relevant academic literature (first of all they have to indicate the existing gap). This point should be enhanced both in the Introduction and in the Conclusions by making reference to the final contribution of the paper.
10) Potential limitations of the paper and the future research are not specified. The Authors have to mention these aspects.
11) Line 187: “which is in line with research by Porter and Guth 5”. Please put 5 in the brackets. I cannot find in the references research done by Guth.
The overall quality of manuscript is good, the paper presents its discussion/argument well. In my opinion, the paper needs to be a bit improved, as indicated in this review. A profound literature review could be performed in order to provide a better context/background. I wish the Author(s) the best of luck with its further development.
Author Response
Comment | Implementated Ammendments |
The aim of the paper should be presented in the Introduction (be consequent, formulate it as it is done in the Abstract). | We added the aim of the study to the introduction (line 69-71) "This study thus aims to investigate the knowledge of HCPs towards the dimension of VBC and to identify potential needs for further education towards implementation of VBC in Germany." |
Line 44: Please use firstly the full name (Value Based Care) in the Introduction and put the abbreviation in brackets (VBC), as it is done in line 21 (Abstract); then you can use the abbreviation in the rest of the paper with no need to explain it. | We changed the part in line 44 to "value-based care (VBC) " |
The theoretical background of the study is not very strong (presented only in the Introduction), please provide additional theoretical support to this study. It could constitute a separate section Literature review/ Literature background. You should include a literature review section after the Introduction and consider formulating research questions or hypotheses. | We added the following paragraph to the introduction and included additional literature (lines 50 to 55): "VBC thus also incentivizes providers to maximize the difference between patient health benefits and the total cost incurred for a service [6]. Due to the necessity to measure outcomes and costs in VBC, the concept is closely linked to evidence-based medicine. Which enables the translation of medical evidence to value measures such as improvement of quality of life through an integration of research evidence, clinical expertise and patient values and [7]." |
Research methods – some additional information about the methodological procedures is necessary, specifically about the hospitals chosen by the Author(s). It is written: “HCPs were contacted directly via their email or their respective intranet (89 German University and non-University hospitals were included)”. Just few phrases would be recommended to characteristic these hospitals. What was the criteria of choosing these 89 hospitals? | We ammended the mentioned part to give additional information: "HCPs were recruited by sending the survey-link to the HCP-mailing list of a big Ger-man private hospital chain, which covers all clinic sizes and thus the entire spectrum of inpatient care in Germany (89 German University and non-University hospitals were included). " line 109-112 |
Maybe because the literature search was performed for articles from the search engine PubMed® of the United States National Library of Medicine, 82.7% of the publications originate from the United States, and only 6.1% from Europe. Perhaps as the paper concerns Germany, it would be reasonable to perform the literature search using another abstract and citation database of peer-reviewed literature. The literature study is quite flat and limited. This section does not bring the reader to a clear state of the art of the literature on this topic. It would be interesting to read about research already conducted in Germany and related to this area. What about other countries? Is it somewhere already done (similar surveys)? Are 556 articles theoretical ones or empirical ones? It would be interesting to know if they present the global perspective or specific solutions for a precise country or even for one HCP (case study), etc. The characteristic of selected articles used in the study is missing. | To explain the reasoning for using PubMed we included the following in the methods part (line 83-85) "The search engine PubMed® was selected as there is no similar platform specifically for the German context. Generally German publication in the field are also represented within this search engine." We added the inclusion criteria more clearly in the methods part (line 81-82): "All articles in English or German language with a study aim clearly related to VBC were included in the analysis." We further included the following in the results section to provide more context on the literature research : "Especially in Europe the amount of research was still very limited in the field at the point of time when the literature review was conducted. However, numbers of publi-cations in general per year increased significantly within the last 10 years as shown in Figure 1. To the authors’ best knowledge no similar survey to the one presented here was conducted in the German context in any other study found in the literature review. " (line 127-132) |
The Authors must justify that they truly fill a gap (theoretical / methodological / empirical) in the relevant academic literature (first of all they have to indicate the existing gap). This point should be enhanced both in the Introduction and in the Conclusions by making reference to the final contribution of the paper. | We included a paragraph in the introduction "By doing so the study delivers an important contribution to the academic literature as well as practice. As it closes the gap in understanding the perspectives of German HCPs on VBC and a creates transparency on pain points that need to be considered in future VBC related policies." (line 71-75) And conclusion (line 292-296): "The present study thus contributed to the existing academic literature by developing an understanding of the perspectives of German HCPs on VBC and creating transpar-ency on pain points that need to be considered in future VBC related policies." |
Potential limitations of the paper and the future research are not specified. The Authors have to mention these aspects. | We added the following paragraph to the discussion (lines 269 to 279):We are aware that the study at hand has certain limitations. Most importantly due to the send-out of the online survey to a pre-selected group, not all potentially in-terested HCPs had access to the survey. Most of the survey participants were medical doctors with an interest in VBC which limits the generalizability of our findings to all HCPs. Generalizability is further limited as no inference analysis has been conducted. Furthermore, the sample of participants displays an underrepresentation of older age groups, which maybe due to the lower affinity to online surveys of certain age groups or due to organizational and hierarchical structures within hospitals. This may lead to skewed results. The choice of a primarily quantitative online survey based on closed questions can furthermore not fully gather and appreciate potential qualitative as-pects of the topic. Another limitation of the study at hand is, that the validity or relia-bility of the questionnaire were not tested |
Line 187: “which is in line with research by Porter and Guth 5”. Please put 5 in the brackets. I cannot find in the references research done by Guth. | The reference [5] was corrected to "Porter ME, Guth C. Redefining German health care: moving to a value-based system: Springer; 2012."; bracktes were included |
